# Performance Assessment of a nZEB Carbon Neutral Living/Office Space and Its Integration into a District Energy-Hub

**Pietro Florio [1,*], Xavier Tendon [1], Jérémy Fleury [1], Carlotta Costantini [2], Andreas Schueler [1] and Jean-Louis Scartezzini [1]**

[1] Solar Energy and Building Physics Laboratory, Ecole Polytechnique Fédérale de Lausanne, CH-1015 Laussane, Switzerland; xavier.tendon@epfl.ch (X.T.); jeremy.fleury@epfl.ch (J.F.); andreas.schueler@epfl.ch (A.S.); jean-louis.scartezzini@epfl.ch (J.-L.S.)

[2] Roechling Automotive, via Nobel 11, 39055 Laives (BZ), Italy; ccostantini@roechling.com

[*] Correspondence: pietro.florio@epfl.ch or pietroflorio@gmail.com

**Abstract:** The integrated performance assessment of buildings can orient their design in the early stages. Despite the wide availability of physics simulation-based, data-driven, and hybrid techniques, it is often difficult to rely on a single, appropriate technique to obtain reliable results. A set of methods, each featuring advantages and limitations, help to refine the performance assessment in an iterative comparative process until a comprehensive picture of the building is achieved. The approach was implemented on a nearly zero-energy building, recently built-up as a combined living and office space (e.g., the SolAce unit) on the NEST infrastructure in Dübendorf (Switzerland). The proposed approach showed that the unit reaches high energy performance accordingly requiring optimal cooling management, involving the control of the opening of blinds and windows. A sound convergence between the computer simulations and data-driven analysis were observed, attesting to the overall energy consumption, of around 26 kWh/m²year, in continuous decrease, aiming at an annual energy-positive balance. The unit was ranked first according to the dynamic energy exchange scheme of the energy trading hub within the NEST facility, which features high-level building modules as a testbed of future building technologies. Embodied energy is estimated at 39 kWh/m²year, which is below the commended limits of Swiss eco-building standards. By considering the carbon sequestration of the wood products during their lifespan, the unit is very close to carbon neutrality with the $CO_2$ emitted annually by the unit over its lifetime being compensated by those stored within wood products during the same period.

**Keywords:** solar architecture; BIPV; energy hub; nZEB; building simulation; energy data

## 1. Highlights

- An innovative combined living/office space is connected to an 'Energy-hub'.
- The unit includes novel bioclimatic developments of the building industry.
- Energy performance is assessed in the concept phase, construction, and operation.
- Used and compared simulation vs data monitoring for energy and comfort assessment.
- High energy performance and neutral $CO_2$ emission; plus-energy exported to the hub.

## 2. Introduction

Despite a considerable increase in energy efficiency, buildings are still accounting for 41% of the final energy demand and 60% of the electricity consumption in the EU-28 [1]. New buildings will play an essential role in reaching the 2050 targets, especially as a

model of excellence, showcasing useful technologies for future renovations. Nearly zero-energy buildings (nZEB), net zero-energy buildings (NZEB), and energy-plus buildings appear worldwide as exemplary energy-conscious building design.

According to Article 2 of the Energy Performance of Buildings Directive of the European Union [2], "nearly Zero-Energy Building means a building that has a very high energy performance. The nearly zero or very low amount of energy required should be covered to a very significant extent from renewable sources, including sources produced on-site or nearby".

World countries have, however, adopted different designations and definitions for these buildings. In the USA, between 2012 and 2019, 580 buildings were certified as net zero and/or energy positive buildings [3].

In Switzerland, during the period 2011–2018, 894 buildings reached the corresponding Minergie-A target, among circa 2 million buildings in the country, representing a useful floor area of more than 600,000 m$^2$ [4]. Other countries focus rather on the concept of nZEB, which is supposed to be more cost effective: in China, between 2010 and 2017, almost 2 million m$^2$ of nZEB floor area were built, and the number of nZEB buildings was expected to reach 5000 units by 2020 [5].

In the European Union, an enquiry in 2014 counted 411 nZEBs [6] and a survey conducted among European architects highlighted that "by 2016 the perceived share of high performing buildings (similar or close to nZEB definitions) in new construction was just above 20% across the EU-28" [7] (p. 84).

Overall, considering that the current global building stock is estimated at more than 220 billion m$^2$, with a yearly growth of 5.5 billion m$^2$, the world will count approximately 415 billion m$^2$ of built area in 2050 [8]. As such, despite the reduction in carbon emissions of the building stock by 30% since 1990 in Switzerland due to efficiency measures and renewables integration [9], the cited efforts seem inadequate to reach the carbon neutrality of the building sector worldwide by 2050, the current NZEB stock being estimated at less than 1% of all buildings on Earth [10] (p. 18).

Despite the progressive simplification and the integration of environmental criteria in the early stages of the project phases, the design effort required to attain high sustainability rates is not affordable to all architects. An enquiry of the Architects' Council of Europe warns that architectural firms dealing with nZEB standards for more than 50% of the time decreased from 14% in 2016 to 11% in 2018 [11]. Dynamic computer simulations are becoming increasingly popular and accessible to help estimate the impact of buildings on the environment. However, their use in the design practice still requires a lot of financial resources and specific skills. Often, the energy performance observed during operation does not correspond to the expectation of the design phase.

In this study, we compare the results of a dynamic energy simulation with the monitored performance of a mixed living and working unit in an innovative research infrastructure in proximity to Zurich, Switzerland. Reducing the difference between the calculated and measured energy demand leads to a progressive convergence towards the real performance. The aim is to check the opportunity of closing the gap between the design phase and operation, towards an effective realization of nZEB buildings.

In this paper, a set of hybrid methods is used to assess the energy performance of the combined living/working space, approaching the quantification problem from both a simulation-based process with incremental effectiveness and a data-driven regression that infers physical data to explain and break down the monitored energy use and production (Figure 1).

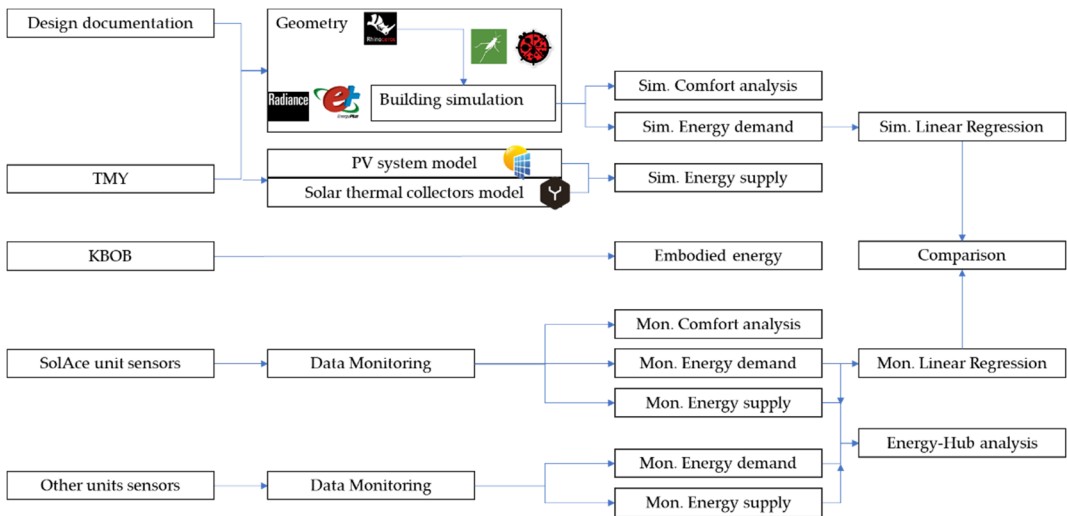

**Figure 1.** Overall methodology workflow.

## 3. Description of the Unit

The NEST infrastructure is a research building located in Dübendorf (Zürich, Switzerland) [12]. New building technologies, materials, and systems can be tested, developed, and validated under real operating conditions. The project involves tight cooperation between academia, industry, and the public sector, whose aim is to ensure that new construction and building technologies are further developed and can reach the market faster. The main feature characterizing the NEST infrastructure is the 'Energy-hub', an innovative platform optimizing energy management from building units to the district level. The 'Energy-hub' comprises many components for harvesting, collecting, storing, and releasing energy flows, connected and managed through a control system. These components can operate individually or be combined.

The different NEST units are connected to various energy distribution networks (i.e., high-temperature water network HTE, medium-temperature water network MTE, low-temperature water network NTE, and electricity network). The energy flow can circulate in both directions, allowing for energy exchange between units through the Energy-hub. The excess heat which is produced during summer is accumulated in a seasonal storage system and released during winter. The hub includes an ice storage facility dedicated to cooling. Photovoltaic modules generate electricity both on a short and on a long term program: the short-term electricity production is stored in batteries, while the long-term one can be converted into either methane or hydrogen. The hydrogen gas is produced through electrolysis using the excess of electricity and consequently stored in special tanks. It can be used either to power fuel cell vehicles or can be channeled back into the building and converted into heat and electricity. Moreover, by using the carbon dioxide in air, hydrogen can be converted into synthetic methane which can feed the gas grid or can be used directly in vehicles.

Several innovative modules of different usages are installed at NEST: the resulting vertical district is offered as a test environment to the Energy-hub. The NEST modules consist of seven independent building units in operation: Vision Wood, Meet2Create, Solar Fitness and Wellness, HiLo, Urban Mining and Recycling, DFAB HOUSE, and SolAce. Vision Wood is dedicated to innovative wood modular construction and Meet2Create is a laboratory of social and sustainable working environments. Solar Fitness and Wellness relies on electricity generation from physical exercise, which also has a wellness area that operates on solar power, while HiLo showcases the possibilities of lightweight construction. Urban Mining and Recycling demonstrates an optimized use of natural resources (waste and recycled materials) by preserving an appealing architectural form. DFAB HOUSE is a residential unit that was built predominantly using robots and

3D printers; SolAce is a mixed living and working space focused on daylighting comfort and envelope prototypes.

The SolAce unit was designed by EPFL researchers to meet the highest standards in terms of indoor comfort and energy performance (Figure 2). Several innovative building technologies were installed in the latter in order to optimize the overall energy balance (Table 1). An intensive use of daylight and passive solar gains is achieved within the unit, besides reaching a positive energy balance over the whole year through 'on-site' generation of solar power and heat. The SolAce unit comprises multifunctional facades involving several novel technologies developed by the research group of Nanotechnology for Solar Energy Conversion at EPFL Solar Energy and Building Physics Laboratory: this includes nanotechnology-based glazing for solar photovoltaic modules [13] and solar thermal collectors [14], innovative micro-structured glazing providing seasonal dynamic management of daylight and solar gains [15,16], and insulating triple glazing with laser-engraved special low emissivity (low-e) coatings for enhanced telecommunication signal transmission [17,18] (Figure 2d). Advanced building sensing and control technologies interact with façade elements and the indoor environment: fostering human–building interaction (HBI), they provide a user-centric approach favoring users' visual and thermal comfort. These include a glare-meter based on a high dynamic range (HDR) light sensor [19,20] which controls the venetian blinds and the electric lighting in the open-space office: the control logic is designed to incorporate thermal, visual, and non-image-forming effects of light. All technologies related to the envelope are complementary to touch all the aspects of building performance, from passive to active solar gains, from tailor-made smart materials to advanced control loops. The SolAce unit combines a working and living space to mitigate the treated floor area for indoor activities and reduce the heating and lighting energy intensities of the unit as well as its environmental impact. Made of prefabricated modular wooden elements and equipped with cardboard-made furniture, the SolAce unit design fosters building materials with low embodied energy and $CO_2$ contents in order to reach carbon neutrality [12,21].

**Table 1.** List of innovative components installed in the unit.

| Light/heat redirecting 3D microstructures (m) |
| :---: |
| Laser-treated selective glazing (lh) |
| Anidolic venetian blind |
| Structural glazing and wood construction |
| HDR vision sensing technology |
| Dynamic and circadian LED lighting |
| Colored glazing for PV and solar thermal collectors (PV and STC) |

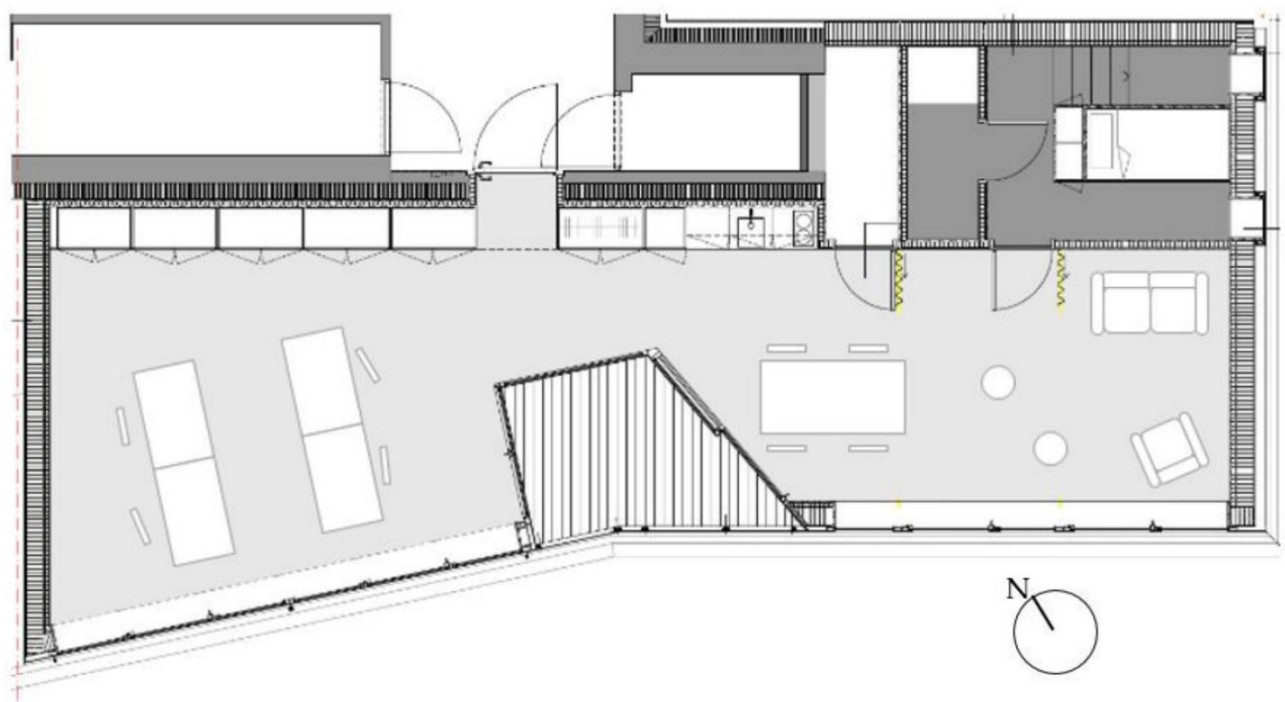

(a)

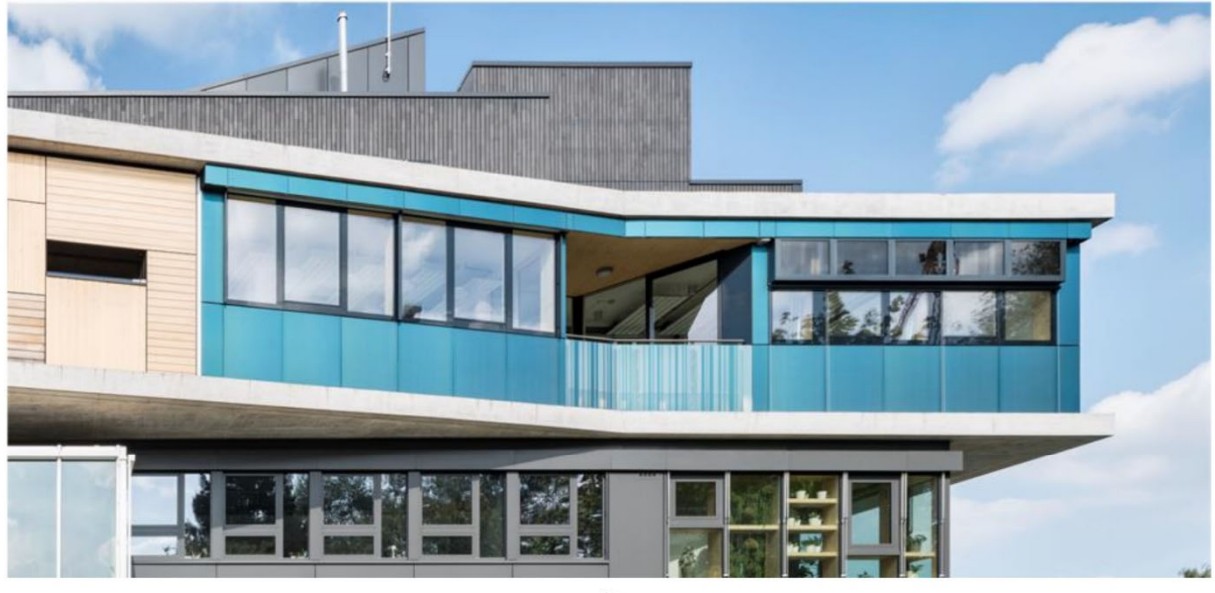

(b)

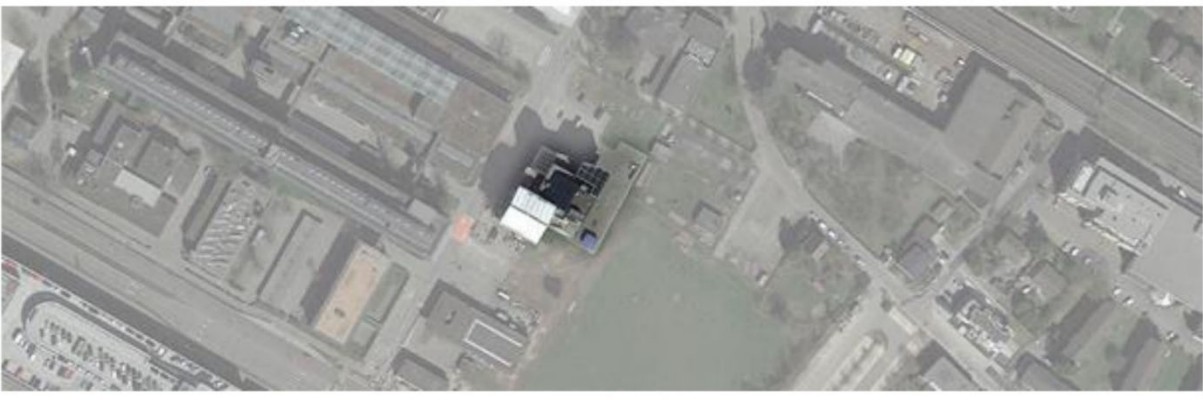

(c)

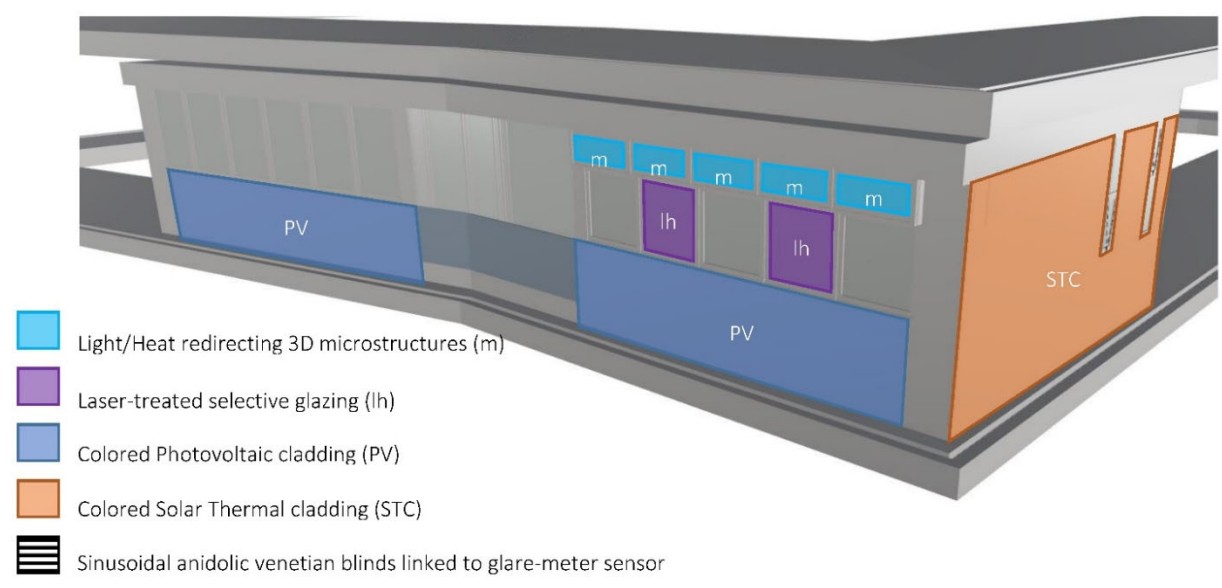

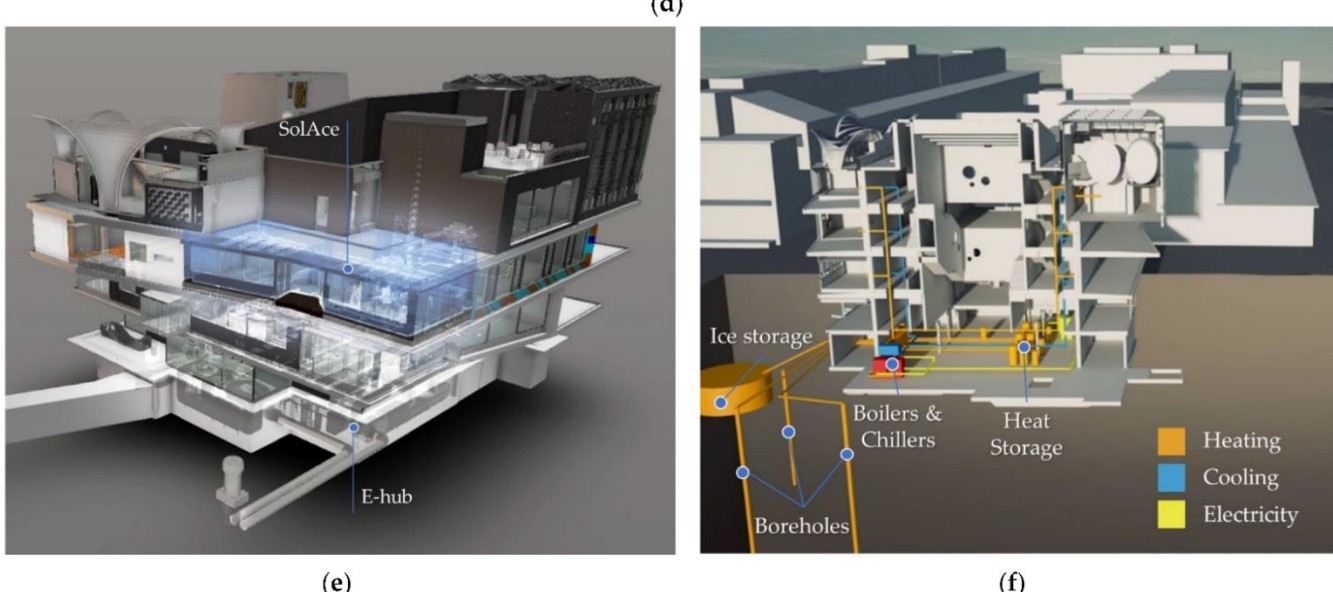

**Figure 2.** Plan (**a**), outdoor view (**b**), multifunctional façade representation (**c**), overview context map (**d**), and context model (**e,f**) of the NEST SolAce unit and the Energy-hub (plan—courtesy of Lutz Architectes, photo—Roman Keller©, context map—Google Earth©, context model of the NEST—Autodesk research©, and context model of the E-Hub—EMPA©).

## 4. Simulation Approach

The simulation methodology relies on different software with a specific purpose for each: (i) Rhinoceros [22] to set up the building geometry, (ii) Grasshopper with Ladybug tools [23] as an interface between geometric data and simulation engines, and (iii) EnergyPlus [24] and (iv) Radiance [25] respectively as energy and lighting computational tools.

The SolAce unit was modeled as a single thermal zone. All façades are exposed to the outdoor air temperature (Table 2), without any weighting coefficient.

**Table 2.** Mean absolute difference between monitored (2018–2019) and typical meteorological year values for outdoor dry bulb temperature and solar radiation, per month.

| Month | Jan. | Feb. | Mar. | Apr. | May | Jun. | Jul. | Aug. | Sep. | Oct. | Nov. | Dec. |
|---|---|---|---|---|---|---|---|---|---|---|---|---|
| Solar Radiation (W/m²) | 41 | 93 | 125 | 123 | 159 | 219 | 194 | 160 | 133 | 72 | 44 | 30 |
| Outdoor temperature (°C) | 4.2 | 4.3 | 4.7 | 6.3 | 3.9 | 6.8 | 5.4 | 5.3 | 4.6 | 3.3 | 4.2 | 5.6 |

The external surface temperature of other interface elements was determined, following the EnergyPlus documentation [26], by weighting the outdoor air temperature by a factor proportional to the buffering effect they generate, and the indoor air temperature by the ratio between the surfaces facing a heated zone and total surface. Since there is currently no heated space above the SolAce unit and the overarching concrete slab is 45 cm thick, the temperature of the outward-facing ceiling $\theta_o$ (°C) is supposed to equal the sum of the outdoor air temperature $\theta_e$ (°C), weighted by a coefficient $N_4 = 0.65$, and the indoor air temperature $\theta_i$ (°C), weighted by a coefficient $N_7 = 0.1$ (Equation (1), excerpt from EnergyPlus documentation [26]). The unit floor is exposed to a heated space with weighting coefficients equal to $N_4 = 0.1$ and $N_7 = 0.8$, since the underneath NEST unit is heated. Other units as well as the central building core, designated as the 'NEST backbone', are supposed to be at a temperature mix assuming $N_4 = 0.4$ and $N_7 = 0.5$.

$$\theta_o = \theta_e \cdot N_4 + \theta_i \cdot N_7, \tag{1}$$

The SolAce heated floor area equals 94.3 m², the net and gross conditioned volumes are equal to 237.8 m³ and to 297.2 m³, respectively. The U-values of the envelope elements are listed below. Window U-values for both glazing and frame, retrieved from the manufacturer's datasheets, are on average equal to 0.95 W/m² K (Table 3), whereas the optical window characteristics are e-coated triple glazing (Table 3). Det
3ails concerning venetian blinds and roller shades are those provided by their manufacturers.

**Table 3.** Thermo-physical characteristics of the SolAce unit envelope

| Envelope Element | U-Value [W/m² K] | Specific Heat Capacity [kJ/m² K] |
|---|---|---|
| Exterior wall | 0.17 | 29 |
| Interior wall | 0.18 | 29 |
| Floor | 0.19 | 61 |
| Ceiling | 0.22 | 44 |
| **Window properties** | | **Value** |
| Glazing U-value | | 0.5 [W/m² K] |
| Frame U-value | | 1.52 [W/m² K] |
| Solar heat gain coefficient (g-value) | | 0.59 [-] |
| Luminous transmission factor (τ-value) | | 0.72 [-] |

Regarding occupancy, the maximum number of occupants is three, with a metabolic heat gain of 120 W/person. The unit occupancy is issued from the US DOE OpenStudio default library assumed for a medium office space [27]. The lighting and electric appliance power densities are equal respectively to 9 W/m² and 1.6 W/m² during peak load conditions. Such values were calculated by summing the connected electric power of each luminaire according to the supplier documentation, and the electric power of a few electronic devices (150 W). Both follow a load profile issued from the US DOE OpenStudio default library for an office building.

Both natural and mechanical ventilation operate in the SolAce unit. The infiltration rate was set to the standard value of 0.15 m³/h m²HFA (SIA 384/2 2020) for airtight constructions, whereas natural ventilation was computed through a simple wind speed and stack effect model. Considering the openable windows distribution, it was assumed

that 43% of the total glazed area can be opened across the entire window height to let fresh air enter the unit. At the scheduled times, three times per day, windows are opened whenever specific temperature conditions triggering natural ventilation are verified: the minimal and the maximal indoor temperature for such openings are 23 °C and 28 °C, respectively, while the corresponding outdoor temperatures are equal to 0 and 26 °C, respectively. The minimal indoor–outdoor temperature difference was set to 5 °C; windows opening is inhibited for wind speeds larger than 40 m/s. As for mechanical ventilation, the total amount of fresh air entering the unit is the sum of two specific airflows, set according to the HVAC technical drawings: (i) the first one is a specific airflow per conditioned area unit, supposed to be set to a minimum of 0.43 $m^3$/h $m^2_{HFA}$ and (ii) the second one is a variable airflow depending on the number of occupants, set to a minimum 36 $m^3$/h pers (ISO 17772 2017). A heat recovery system with 40% effectiveness is included in the double-flux mechanical ventilation system.

In the current simulation, a daylight-responsive control strategy was adopted to manage the operation of shadings. The control point is located at a height of 120 cm above the floor in the center of the open space office room, with a view facing the terrace. If the daylight glare index (DGI) exceeds the threshold limit between comfort and discomfort of 22, the blinds are entirely deployed. The blinds slanted angle is constant and equal to 25 degrees downwards from the horizontal plane.

The back-up heating system was modeled as an ideal air load system [28], supplying air at the desired temperature set point with a 100% ideal efficiency: the model outcome is similar to the useful energy demand monitored in the unit. The heating and cooling peak power of the system were set to 43 $W/m^2$ and 22 $W/m^2$, respectively, according to the specifications of the HVAC retrieved from technical documents. The heating set point temperature was set to a constant value of 22 °C, as observed during monitoring, in order to meet thermal comfort in the unit at any time. However, the switching on of the back-up heater is triggered by outdoor air conditions: heating is allowed for a 24 h period if the mean outdoor air temperature is lower than 11 °C, it is switched off for a 24 h period if the mean outdoor air temperature is above 12 °C. A similar situation occurs for the cooling operation, given a cooling set point of 22 °C, as observed during monitoring, which is only activated for an average outdoor temperature higher than 15 °C within 24 h before and is triggered off if the temperature reaches lower than 14 °C (Table 4).

**Table 4.** System operation characteristics in the SolAce unit.

| Feature | Value |
| --- | --- |
| Heating set point | 22 °C |
| Cooling set point | 22 °C |
| Heating on | Daily mean outdoor air temperature < 11 °C |
| Heating off | Daily mean outdoor air temperature > 12 °C |
| Cooling on | Daily mean outdoor air temperature > 15 °C |
| Cooling off | Daily mean outdoor air temperature < 14 °C |

The meteorological data set was generated using Meteonorm software in the form of a typical meteorological year (*TMY*), a weather representative data set built-up using average values for all months based on the latest decades [29]. In this specific case, temperature data belong to the historical series 1991–2009 and solar radiation data to the historical series 2001–2009 (Figure 3).

The geometric shape of the surrounding buildings was retrieved from aerial photography as well as from a rough estimation of their height based on the number of floors. Shading calculations from the surroundings are updated every single day at an hourly time step. The distribution algorithm of solar radiation in the unit assumes that the whole solar beam enters in the thermal zone and impinges on the floor. The terrain

roughness affecting the wind speed around the NEST building as well as the external heat losses is typical of urban conditions (e.g., set to suburbs in EnergyPlus).

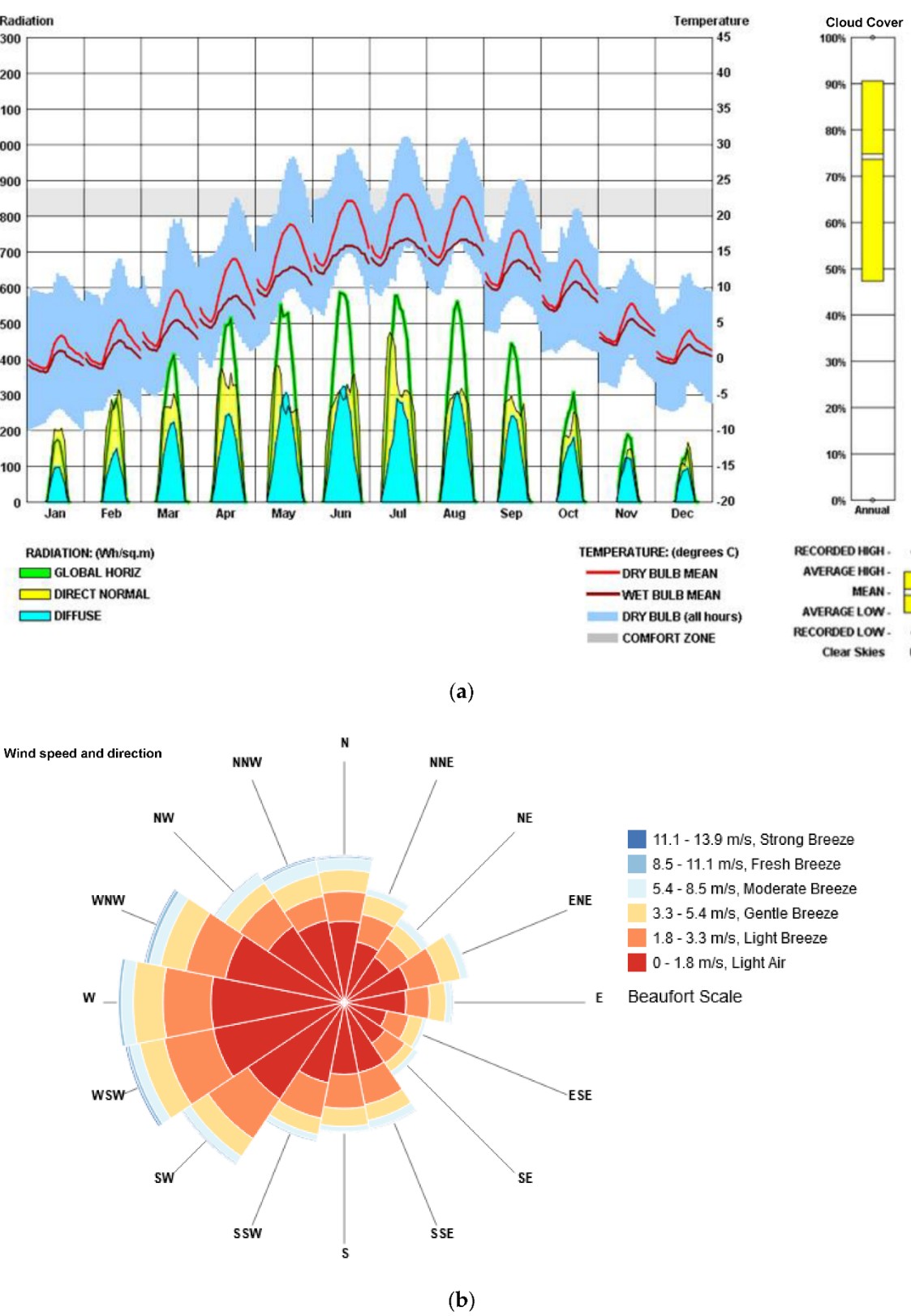

(**a**)

(**b**)

**Figure 3.** Climate context, extracted from the typical meteorological year (*TMY*). Monthly radiation (left axis) and temperature (right axis) in the left figure, cloud coverage in the right figure (**a**), and wind speed and direction (**b**).

SolAce can be used as a workspace or as a living unit: this makes the modelling of domestic hot water (DHW) needs more complicated. The DHW demand is also significantly affected both by user behavior and unit occupancy. The hot water consumed volume is estimated at 40 L/day per person, in accordance with the standards (SIA 385/2 2015). The DHW production requires 10.4 kWh/m²$_{HFA}$ year for an average presence of a single person per day. This comprises the storage losses, a hot water temperature of 55 °C, a cold water temperature of 15 °C, and a storage temperature of 50 °C. All the domestic hot water used in the unit is considered to leave the related thermal zone with its overall calorific content.

The SolAce unit is equipped with solar thermal collectors and a 150 L storage tank to cover a fraction of the DHW needs with an energy conversion efficiency of 58%. The collectors are located on the southeastern facade of the unit, oriented with an azimuth of 119° with respect to the geographical north. They cover a gross area of 19.5 m² and show a net collector area of 18 m². The simulation of the solar thermal production was carried out using the POLYSUN software tool [30]. The Energy-hub backs up the solar thermal system, as an auxiliary hot water producer, due to a dedicated water loop that operates at a temperature range of 45/60° C. The latter acts both as a heat source and/or a heat sink, with the heat flow direction being regulated by a six-way valve, opportunely controlled.

Renewable electricity is provided by eleven solar photovoltaic modules which are integrated on the southwestern facade of the SolAce unit. Their total gross area equals 17.1 m², their net area 12.3 m², and their peak power reaches 2.1 kWp. Modules are characterized by a conversion efficiency of 17.3% at standard temperature conditions (STC) and an annual performance ratio of 83.1%. The PV solar system was simulated using PVSyst software [31].

Finally, a life cycle assessment (LCA) was conducted to determine the embodied energy and carbon emissions due to construction and disposal of the unit beyond its operational energy demand [32]. The KBOB Swiss database for sustainable construction was used for this purpose [33], given the material inventory of the SolAce unit.

## 5. Data Monitoring Approach

In this section, a hybrid methodology to source physical parameters from monitored data via regression techniques is explained; although it is not a "black-box" model as there is no prediction from previously labelled data, it can be qualified as a data-driven approach, given the process which relies on experimental monitoring.

The NEST SolAce unit is fitted with 55 sensors, embedding 284 data points connected to a building automation network, which allows the monitoring of physical, environmental, and indoor comfort variables as well as a continuous control of energy systems (Supplementary Materials—Table S1). Monitored data are stored in a database with a 1 min time step and can be queried through a repository accessible via most coding languages. An online dashboard is also available for visualization and diagnostics; all data mentioned in this section were retrieved through such platforms and downsampled to an hourly resolution.

A regression analysis was carried out considering 283 days of monitoring in total (164 days in summer and 119 in winter). The heating season ranges from 8 October 2018 to 17 April 2019, while the cooling season covers the complementary period of the year 2018–2019. Roughly 15% of the monitored period was excluded due to invalid or missing data or to temporary data acquisition system maintenance. Compared with the typical meteorological year, a slight difference was observed in monitored outdoor dry bulb temperature and in global horizontal radiation, in the order of 5 °C and 100 W/m², respectively, especially in summer months (Table 2).

The heating and cooling energy flows delivered to the unit (useful energy) are known at each timestep of the monitored period through the combination of supply water flowrates and temperature differences. The heat transfer through the glazing is obtained from the $U_{tr}$ value [W/m² K] specified by the window manufacturer (Table 3) and the

respective area of the windows $A_{tr}$ [m²], considering the monitored difference between indoor and outdoor air temperature $\Delta T$ [K]. Hereby, the thermal inertia of the glazing can be neglected. As the heating (or cooling) useful supply power is known, the heat losses (or gains) through the opaque part of the envelope $P_{op}$ [W] can be finally inferred by subtraction of the transparent heat flux from the useful supply power because of energy conservation over the observation period of approximately six months: as such, the heat flux through the opaque envelopes includes the inertial effect of thermal mass. The conduction heat flux is then calculated as follows (Equation (2)):

$$P_{conduction} = (U_{tr} \cdot A_{tr} + P_{op}) \cdot \Delta T, \tag{2}$$

The convective losses $P_{convection}$ [W] can be determined using Equation (3):

$$P_{convection} = D_{air} \cdot c_p \cdot \rho \cdot \Delta T \cdot \eta_r, \tag{3}$$

where $D_{air}$ [m³/h] is the volumetric airflow rate [34], which has been assumed to be equal to half the nominal air flow rate of the mechanical ventilation system (i.e., 180 m³/h), according to the monitored fan operation rate, $c_p$ [J/kgK] is the air specific heat capacity equal to 1004 [J/kgK], $\rho$ [kg/m³] is the air density equal to 1.204 [kg/m³], $\Delta T$ [K] is the indoor–outdoor temperature difference, and $\eta_r$ [-] is the heat recovery effectiveness of the ventilation system. A double flow heat recovery system with an effectiveness of 40% has been accounted for in the computer simulation.

The passive solar gains $P_{solar}$ [W] can be obtained using Equation (4):

$$P_{solar} = A_{tr} \cdot g \cdot I_{GH,m} \cdot \frac{I_{GSW,TRY}}{I_{GH,TRY}}, \tag{4}$$

where $A_{tr}$ [m²] is the area of the fenestration located on the NEST SolAce unit façades and g [-] is the corresponding solar heat gain coefficient. As the vertical solar radiation on the southwest façade $I_{GSW,m}$ [W/m²] was not available from the local meteorological station, a typical meteorological year (*TMY*) for the period 1991–2010 was used. Through the latter, it was possible to determine the ratio between the southwest vertical solar radiation $I_{GSW,TMY}$ [W/m²] and the global horizontal solar radiation $I_{GH,TMY}$ [W/m²] on a daily basis. The southwest vertical global radiation was then calculated with respect to the measured horizontal global radiation $I_{GH,m}$ [W/m²], available from the on-site meteorological station, using the *TMY* available ratio. The solar heat gain coefficient g [-] was determined using the cooling energy needs monitored over six days characterized by an average daily indoor–outdoor temperature difference equal to zero. During such days, the total cooling energy is supposed to perfectly balance the window transmitted solar radiation, given the absence of conductive losses. The passive solar gains effectively penetrating in the unit on a given day divided by the incident solar radiation on the façade constitute the g-value, which is equal to 0.41.

The internal heat gains $P_{internal\,gains}$ [W] are expressed by Equation (5):

$$P_{internal\,gains} = P_m \cdot N + P_{elec} \cdot F, \tag{5}$$

where $P_m$ [W] equal to 120 W is the metabolic rate of a person for a common office activity [34], and $N$ [-] the number of people. The latter has been inferred directly from the measured occupancy data, and by assuming three occupants in the unit when presence is marked. Overall, occupancy is low compared with standards given that the SolAce unit is currently used for research purposes and sporadically as a residential space. $P_{elec}$ [W] is the electric power absorbed by the electric appliance, and $F$ [-] is the ratio that represents the amount of electricity converted into heat. In this case, $P_{elec}$ was inferred from the electrical energy consumed in the observed period and $F$ was assumed equal to 1, for simplicity.

The domestic hot water (DHW) energy demand is issued from the monitoring of the heat supplied by the NEST 'Energy-hub' to the SolAce unit. The same energy flow meter measures the heat delivered to the 'Energy-hub' by the solar thermal collectors located on

the facade of the unit. Currently, there is no monitoring of the amount of heat supplied directly by the solar collectors to the hot water storage tank; as such, the whole excess amount of solar heat is assumed to be delivered and stored for a later use in the 'Energy-hub', independently from its final use. The thermal losses of the DHW storage are determined by calculating the resulting value to close the heat balance of the storage tank. Finally, all data regarding the electric appliances, including the solar electricity production from the photovoltaic modules, are monitored using electricity meters opportunely installed in the circuit.

## 6. Energy and Comfort Analysis

Four Sankey diagrams pertaining to the computer simulations and the data-driven scenarios for the heating and cooling seasons were developed (Figures 4 and 5); data-driven results include an error of 5% in the energy balance. Although the simulation results and the data monitoring analyses turned out to be quite similar, some relevant differences were observed. In winter, the simulation features a solar thermal production equal to 4.4 kWh/day, while the corresponding outcome from the data-driven approach is equal to 10.7 kWh/day. However, this is due to the inaccuracy of the solar thermal model, as well as by the distortion of the meteorological variables due to the use of a *TMY*. Another difference can also be observed for the metabolic heat gains: the simulation returns 2.7 kWh/day, while the monitoring analysis gathers 0.4 kWh/day. This discrepancy can be explained by the constant number of occupants (e.g., three people all day long) assumed in the model. The data monitoring, instead, relies on the presence sensor, with three occupants being assumed if a presence is signaled. The simulated DHW demand is equal to 2.7 kWh/day, with the monitored one indicating 3.0 kWh/day. During the heating season, the conductive losses are larger due to a larger temperature difference correlated with a lower ambient air temperature; the thermal losses are compensated either by energy flows from the 'Energy-hub' or the solar thermal collectors.

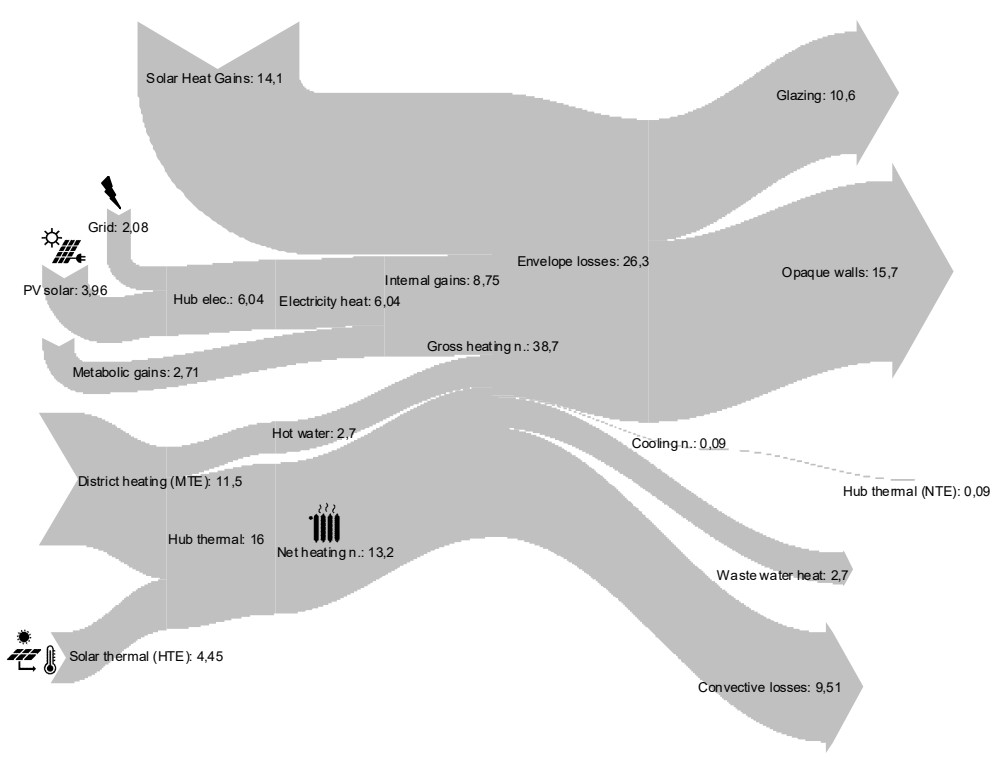

(**a**)

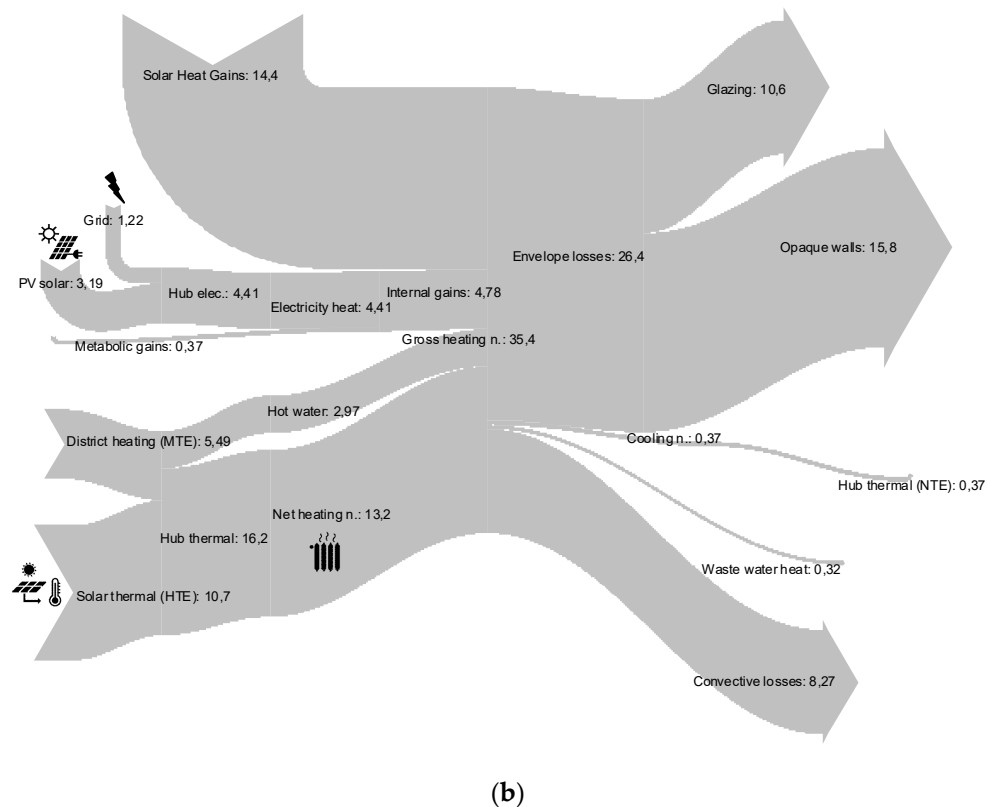

(**b**)

**Figure 4.** Sankey diagrams of the simulated (**a**) vs. monitored (**b**) scenario for the heating season (kWh/day).

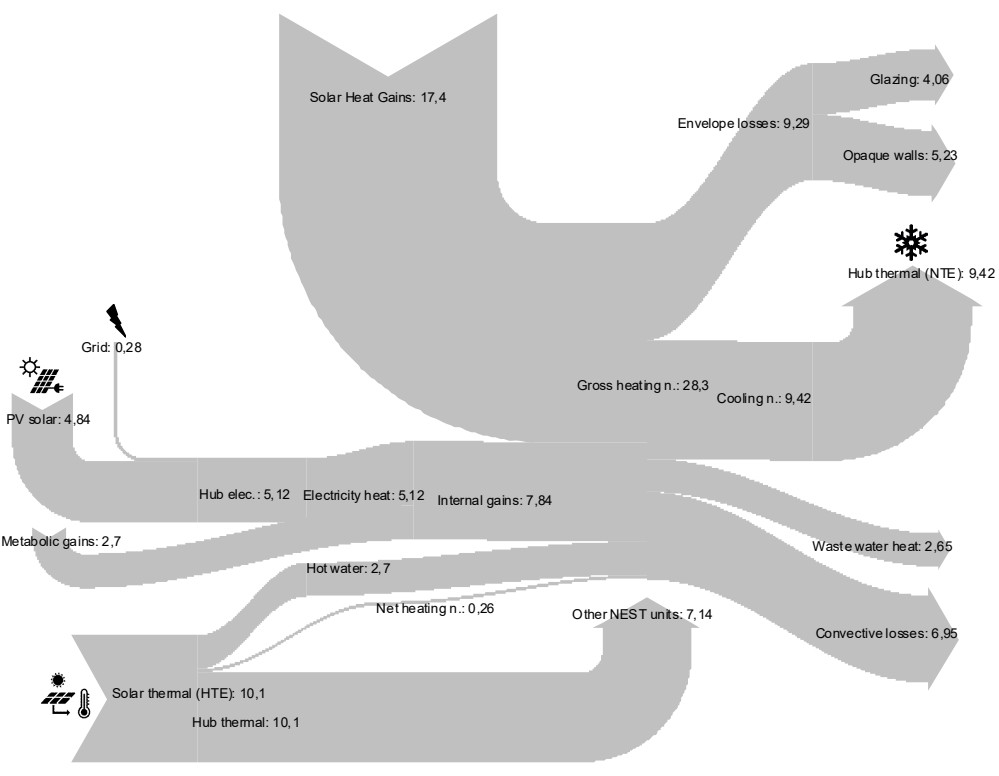

(**a**)

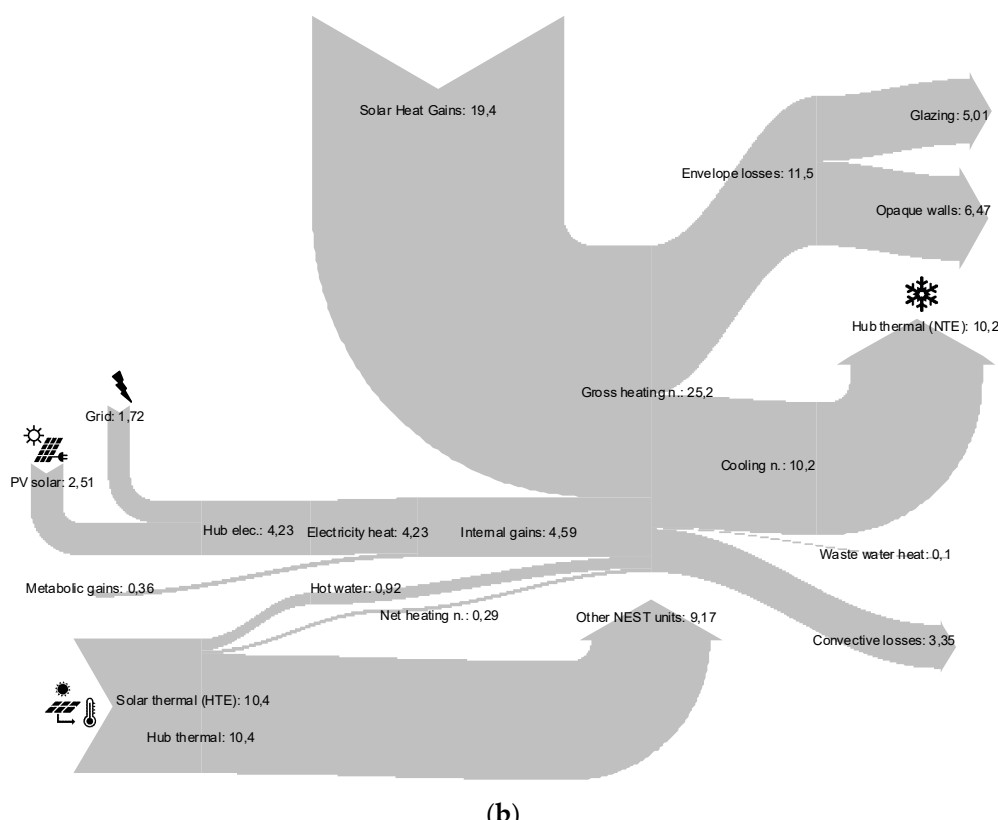

**Figure 5.** Sankey diagrams of the simulated (**a**) vs. monitored (**b**) scenario for the cooling season (kWh/day).

The cooling season presents a difference in the PV solar modules' electricity production: the simulated value is equal to 4.8 kWh/day, while the monitored one is 2.5 kWh/day. The real production is lower due to a large scaffold recently installed in front of the southwestern facade, for the purpose of further construction on the NEST infrastructure, casting shadows on the PV solar modules. However, the electricity production issued from the PV solar collectors on the southeastern facade is not significantly impacted; no shadowing is observed. The monitored DHW demand is lower than the simulated one, since the actual average occupancy of the unit was below one person per day, almost matching the sole hot water storage losses. The envelope thermal losses issued from the simulation are equal to 9.3 kWh/day; the monitored ones to 11.5 kWh/day. Cooling needs reach 9.4 kWh/day in the simulated scenario versus 10.2 kWh/day in the monitored one: a possible reason resides in the windows opening and/or night cooling strategies which are schedule-based in the simulation and may not replicate the actual occupants' behavior in the unit. In the data-driven scenario, all loads are compensated by the cooling system and seldomly released through window openings, given the protracted absence of occupants.

Overall, the unit is characterized by an annual heating intensity of about 30 kWh/m² year (end energy, Figure 6), which is remarkable considering that it might be assimilated to an apartment without any roof surface available for solar energy collection. Solar heat gains are certainly large but well managed, e.g., exploited or rejected, given the need. The total embodied energy amounts to 39 kWh/m² year, assuming a lifetime of 60 years for structural elements and of 30 years for non-structural elements (SIA 2032: 2020, Annex E). Such a result is due to the large use of wood and to the choice of natural resins for floor finishing (Figure 7).

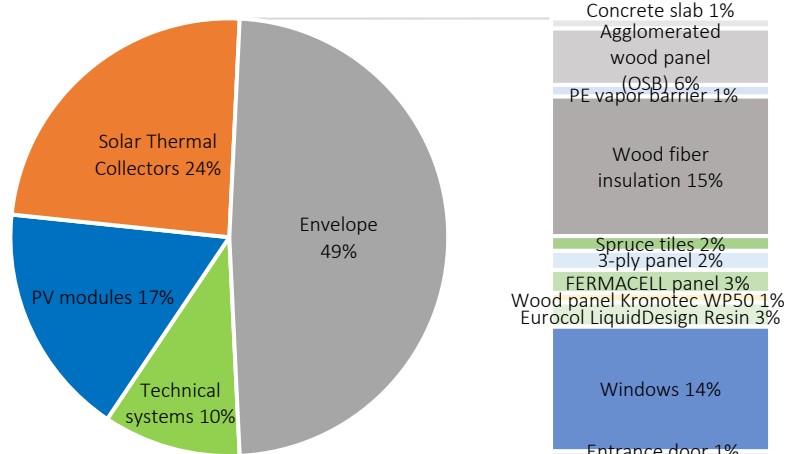

**Figure 6.** Bar chart of the annual end energy consumption vs the production and simulation scenario (**a**) and data-driven scenario (**b**). 'Passive solar gains' are annual and shown only for comparison; $CO_2$ emissions of the Swiss average buildings in 2005, 2050, and for the SolAce unit (**c**).

**Figure 7.** Life cycle assessment of the SolAce unit showing breakdown of different components' contributions to embodied energy use as well as $CO_2$ emissions.

The unit is comfortable for most of the year as outlined by Figure 8, given the system operation characteristics listed in Table 4. Unfortunately, 172 uncomfortable hours (2%)

are indicated by the simulation carried out using the *TMY*, whereas 1017 h (13%) are marked uncomfortable during the monitored period of October 2018 to October 2019. Comfortable conditions are those acceptable by 90% of a standard audience (e.g., a 10% PPD) for a typical office metabolic rate ranging from 1 to 1.2 [met] and a clothing level from 0.5 to 1.2 [clo], up to 0.012 [kg water/kg air] absolute humidity [34]. Some of the difference between the simulated and the data-driven scenarios is explained by the difference between the *TMY* meteorological data and those of the year 2018–2019 (Table 2), a shift towards warmer ambient air temperatures being observed in both cases.

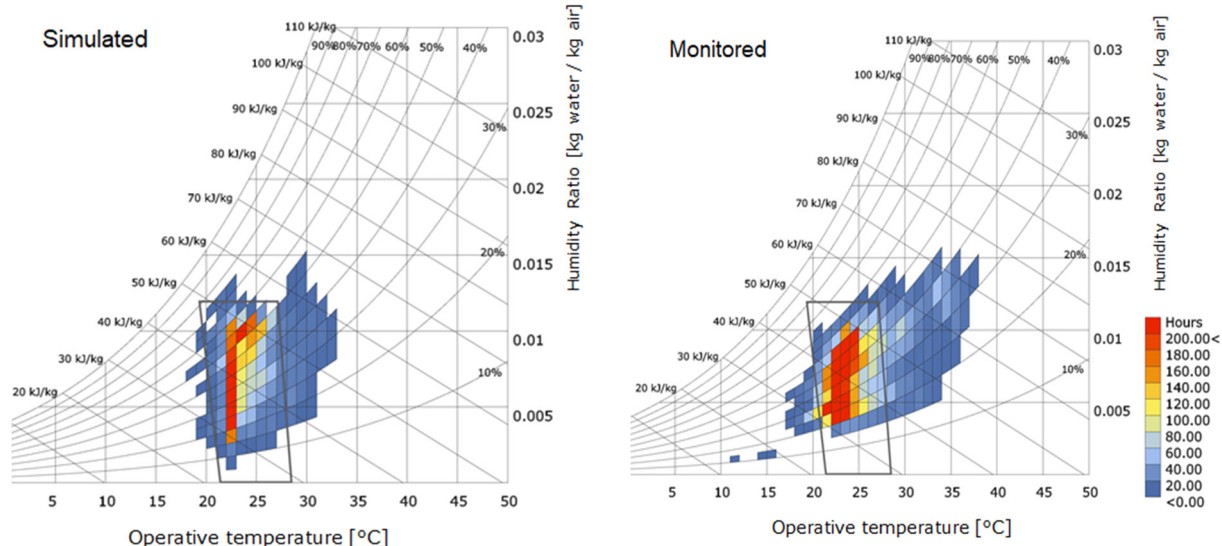

**Figure 8.** Yearly psychrometric diagrams for the simulated (**left**) and the monitored scenario (**right**). The comfort polygon is highlighted in the center of the diagram.

## 7. Energy-Hub

An interesting perspective regarding the energy performance of the SolAce unit within the NEST infrastructure is highlighted by the energy flow exchange between the different NEST units. All units at NEST are connected to the 'Energy-hub' which constitutes an energy storage and dispatching platform. As such, units can extract or deliver heat or electricity (not mass) from or to the 'Energy-hub', which acts as a backup energy source or energy sink accordingly. By imagining a sort of 'peer-to-peer' network, units that are producing more energy than needed at a given timestep, can virtually send their overproduction to units characterized by a larger demand through the 'Energy-hub'. The hourly simulation of this behavior, based on energy supply and demand monitored data for the analysis period (October 2018–October 2019), is illustrated in Figure 9. It is assumed that, besides its own load match, each unit shares the energy surplus equally among other units and finally delivers the eventual remaining energy amount to the NEST 'Energy-hub'. Being characterized by a relatively low annual total demand, SolAce provides several tens of kWh electricity produced by the BiPV solar facade to other units, as well as an amount that exceeds a thousand kWh of high-temperature heat from the solar thermal collectors on the southeastern facade to the 'Energy-hub'. Besides SolAce, the units "Digital Fabrication" and "Solar Fitness and Wellness" produce a considerable amount of electricity due to their location on the NEST rooftop: they also consume a lot. In particular, the "Solar Fitness and Wellness" unit is fitted with heat-pumps which use solar electricity to invert heat flows, thus generating warmth and coldness in the unit. The "Meet to Create" unit being the most occupied throughout the year, shows a higher energy demand.

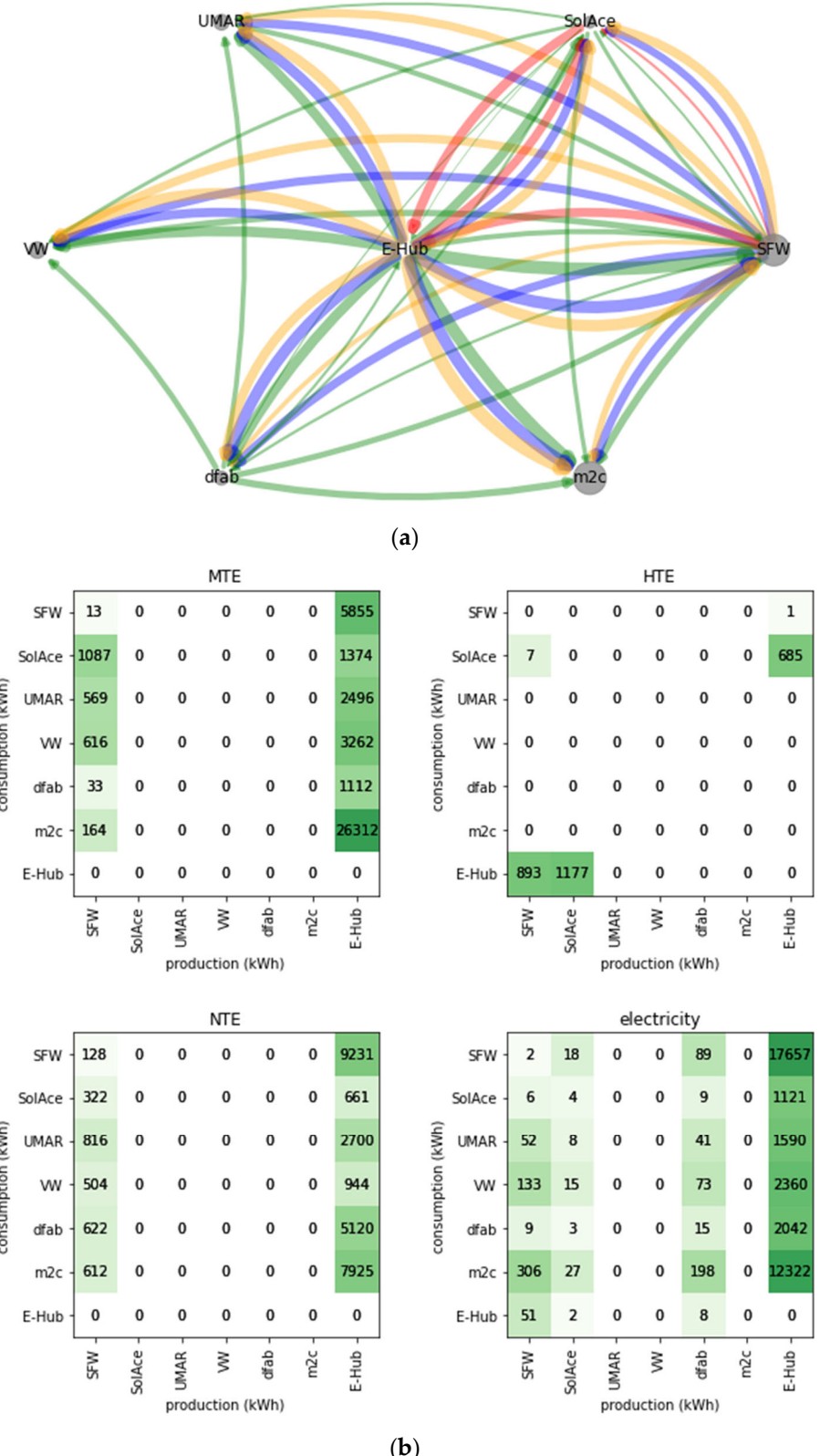

(**a**)

(**b**)

**Figure 9.** Energy exchange diagram between units at NEST (**a**) and relative association matrixes (**b**). In the diagram, flow thickness is proportional to the logarithm of the energy amount, while the node size is proportional to the annual total energy demand of the node.

Estimating the overall performance of the units can be intricate: a rewarding mechanism was fortunately established by the NEST technical staff for sake of comparison (Table 5) [35,36]. Equation (6) [36], (p. 224) shows the calculation of the overall net energy performance figure NEPF [kWh/year] as defined in [36]: it is a function of the different final energy demand and supply flows *Q* [kWh/year], weighted by an exergy-based factor w [-] linked with each energy sub-system (i.e., high temperature water network HTE, medium temperature water network MTE, low temperature water network NTE, and electricity network). The factor w [-] is then corrected according to the season of the year/day of the week the energy flux refers to (see details in [36]), to reward energy fluxes outside of peak time. Table 6 confirms that SolAce is the most rewarded unit in terms of energy exchange, due to the large production of high-temperature heat, especially during summertime (Figure 5). The negative score indicates that energy neutrality is not reached yet, but SolAce is not far (i.e., -2461 kWh/year, namely, 26 kWh/m² year) and has the potential to optimize its performance as highlighted in the following section.

$$NEPF = w_i \cdot Q_{heating} + w_j \cdot Q_{DHW} + w_k \cdot Q_{cooling} + w_l \cdot Q_{PV} + w_m \cdot Q_{STC} + w_n \cdot Q_{elec,} \tag{6}$$

**Table 5.** Rewarding scheme adopted for the NEST units issued from [35], weighting factors for energy flows.

| | Heat Heating Season | | | Heat Mid-Season | | | Heat Cooling Season | | | Electricity Heating Season | | Electricity Mid-Season | | Electricity Cooling Season | |
| --- | --- | --- | --- | --- | --- | --- | --- | --- | --- | --- | --- | --- | --- | --- | --- |
| | HTE | MTE | NTE | HTE | MTE | NTE | HTE | MTE | NTE | week | w.end | week | w.end | week | w.end |
| **Supply** | 0.432 | 0.240 | 0.034 | 0.306 | 0.107 | 0.029 | 0.137 | −0.025 | −0.123 | 1.222 | 0.838 | 1.045 | 0.623 | 0.868 | 0.408 |
| **Demand** | −0.345 | −0.228 | −0.034 | −0.244 | −0.101 | 0.029 | −0.137 | 0.022 | 0.123 | −1.222 | −0.838 | −1.045 | −0.623 | −0.868 | −0.408 |

**Table 6.** Ranking of the units based on the rewarding score, in the analysis period (October 2018–October 2019).

| Unit | Score |
| --- | --- |
| **SolAce** | −2461 |
| **UMAR** | −3163 |
| **VW** | −3759 |
| **dfab** | −6488 |
| **m2c** | −18,289 |
| **SFW** | −26,605 |

## 8. Discussion

After completion of the data collection for the whole analysis period (i.e., October 2018 to October 2019), an assessment of the correspondence with the simulated data was carried out. In particular, the fitting of mono and multi-dimensional linear regressions through the data was conducted.

Figure 10 represents the scatter plot of the energy used both for space heating (red dots) and for cooling (blue dots) at hourly timesteps for the NEST SolAce unit during the analysis period, as a function of the indoor–outdoor temperature difference. The point clouds lead to a regression through a straight line, pointing out a linear dependency between the energy demand and the temperature difference. This dependency confirms the linear relationship between the heating power and the temperature difference assimilated to the energy signature. Cooling needs were assumed as negative.

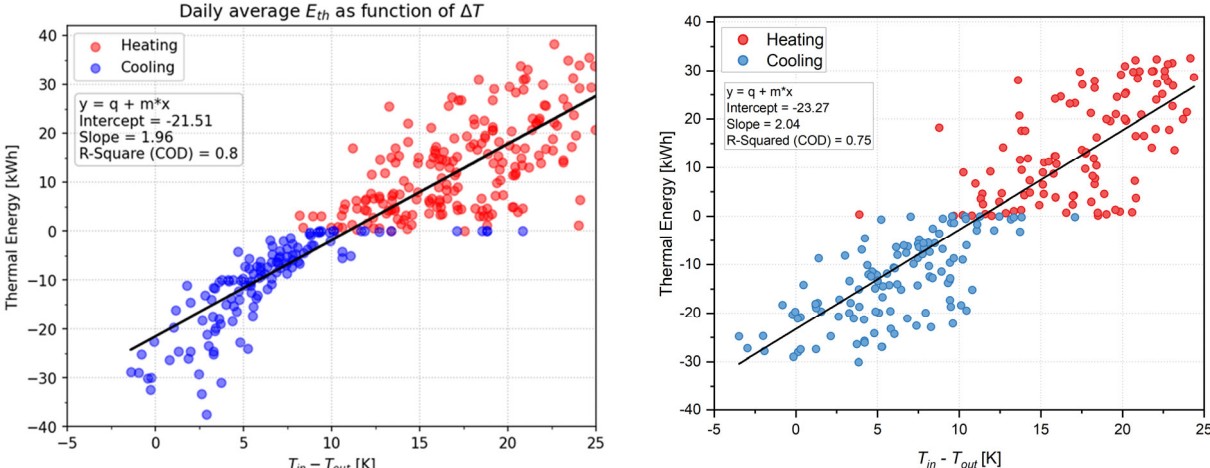

**Figure 10.** Daily heating and cooling demands as a function of average daily temperature difference for simulated (**left**) and monitored data (**right**).

Figure 11 shows the average hourly heating and cooling demands as a function of the global horizontal solar radiation and the indoor–outdoor temperature difference. The regression plane provides more information regarding the main heat sources and sinks. The NEST SolAce unit shows a large window-to-wall ratio (WWR), solar radiation being the main cooling load, especially during interseason and summertime. On the other hand, during wintertime, heat losses through the envelope have a large impact on the heating demand.

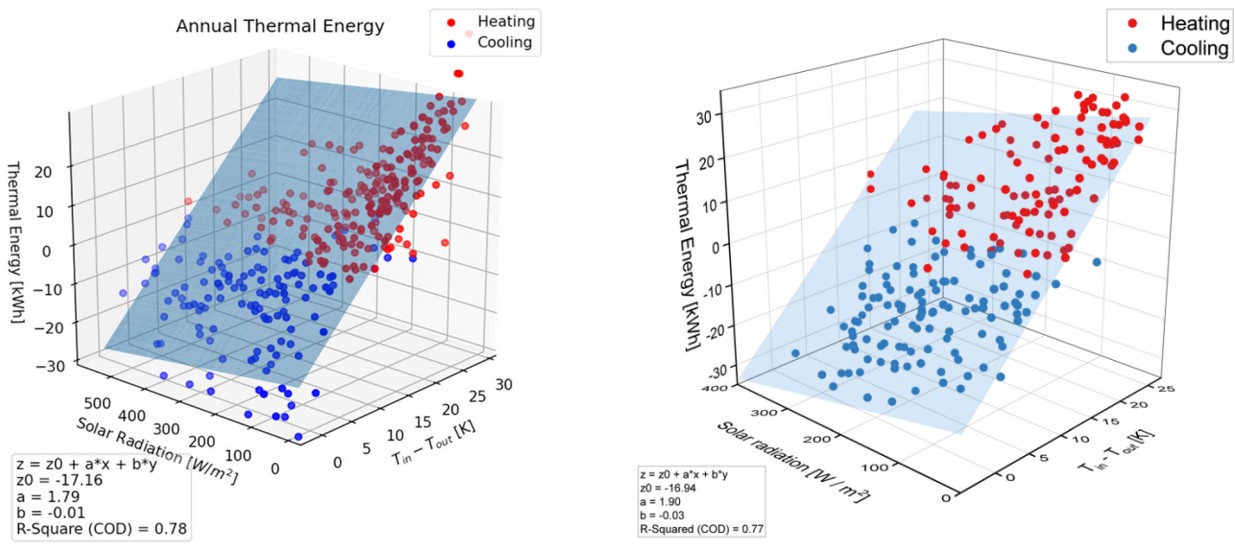

**Figure 11.** Daily heating and cooling demands as a function of average daily temperature difference and solar radiation on the southwestern facade for simulated (**left**) and monitored data (**right**).

Overall, the SolAce unit performs reasonably well from an energy performance perspective (Figure 6). This is confirmed by the compliance with existing voluntary energy labels: the heating intensity is way below the threshold of the Minergie standard [4,37], i.e., end energy $Q_h$ = 30 kWh/m² year against 52 for the Minergie upper limit [37]. The same goes for the primary energy index (MKZ), which is estimated at 30 kWh/m² year compared with the limit of 35 for the more stringent Minergie-A administrative building category. The total primary energy intensity for heating, cooling, ventilation, and DHW production is very close to the limit for all Minergie labels, i.e., 42 kWh/m² year against 40

for administrative buildings. A primary energy factor of 0.4 [-] was considered, given the highly renewable energy mix feeding the 'Energy-hub', as well as an annual energy efficiency of 0.85 [-] for the gas boiler according to the Minergie specifications. By optimizing the cooling strategy, the Minergie conditions could very likely be satisfied. The most demanding constraint for Minergie-A is the need for the (primary) weighted PV solar electricity production to be larger than the primary energy index MKZ: surprisingly, the Minergie standard does not include the solar thermal energy production in the estimation of the renewables' generation. As such, SolAce could not qualify for Minergie-A, given the weighted PV solar production of 33 kWh/m² year according to the monitoring. However, if the weighted solar thermal production was added, the renewable energy amount would be enough to compensate for the overall energy demand, according to the Minergie standard. Regarding its LCA, the SolAce unit features an embodied energy value that is 16% underneath the lower limit of the Minergie-ECO label, namely, an excellent result. In relation to carbon sequestration, the wood products compensate the carbon emissions of the unit estimated from the KBOB [33]. Equation (7) allows the estimation of carbon sequestration:

$$m_{CO_2} = \frac{m_{dry\,wood} \cdot c \cdot f_{CO_2}}{n_{years}}, \tag{7}$$

where $m_{CO_2}$ [kg] is the mass of sequestered $CO_2$ during the product lifespan, $m_{dry\,wood}$ [kg] is the mass of dry wood constituting the product, $c$ [%] is the fraction of dry mass represented by carbon (assumed equal to 50%), $f_{CO_2}$ [kg/kg] is the amount of carbon dioxide cumulated in a unit of carbon mass (considered equal to 3.67 [kg/kg]). Finally, $n_{years}$ [years] is the number of years assumed for the unit life cycle (60 years for structural elements and 30 years for non-structural elements (SIA 2032: 2020, Annex E)). Overall, the 820 kg $CO_2$ emitted annually by the unit over its lifetime is compensated by the 745 kg $CO_2$/year stored within wood products during the same period.

Summer comfort is currently an issue for the SolAce unit, according to the on-site experimental monitoring of the unit, illustrated by Figure 8. The data-driven analysis clearly shows that the comfort boundaries are exceeded during a certain period. Minergie standards allow a maximum of 100 h above the comfort limits during the cooling period based on the indoor and outdoor air temperatures: the SolAce unit overpasses this threshold by 26%.

However, the thermal comfort analysis carried-out by computer simulation using a *TMY* shows optimal comfortable conditions during summertime. The difference can partially be explained by the difference between the weather conditions of the *TMY* data and those of the year 2018–2019. The regular presence of occupants assumed in the simulation model is another main cause of discrepancy: in reality, the occupation rate of the unit was very low during the period 2018–2019. In the on-site experimental situation, windows were closed during unoccupied hours, the monitored indoor temperature rising significantly during the latter. The computer simulations demonstrated that by implementing a blinds control strategy based on the indoor temperature, in parallel to the one set only for glare protection, more than 4 kWh/m² year of energy would be saved for cooling and 2% more comfortable hours would be gained. Together with other correction measures, such as the addition of PCM materials, the installation of new shadings to protect glazing against the terrace (see Figure 2), as well as an automated opening of windows at night (night cooling), savings would reach 10 kWh/m² year on cooling energy and thermal comfort would always be reached.

## 9. Conclusions

A methodology to assess the integrated performance of a combined living/office space (e.g., NEST SolAce unit), designed as a future nZEB and carbon neutral urban apartment, is suggested with clear steps subdivided by phases. Dynamic simulations and on-site data monitoring were used to obtain a reliable energy balance in the form of Sankey

diagrams. After comparison with the energy simulation model, the main variables impacting the energy demand were identified. Overall, the unit is energy positive, when considering the space heating, domestic hot water and electric appliances demand, a remarkable achievement for a combined office/living space. By using a multi-functional facade equipped with complementary energy-optimization technologies and without any roof surface, SolAce provides renewable electricity and high-temperature heat to the NEST energy exchange platform, i.e., the 'Energy-hub', featuring the highest ranking within the rewarding scheme. However, due to its high energy performance (26 kWh/m²year), the unit requires optimal cooling management, involving the control of blind and window opening. Simulations demonstrated that strategies such as automated blinds control and night cooling can mitigate up to ⅔ of the current cooling loads and guarantee a more comfortable space to occupants. Made of prefabricated modular wooden elements and equipped with cardboard-made furniture, the SolAce unit uses materials with a low embodied energy and carbon content. Embodied energy is estimated at 40 kWh/m²year, which is below the Minergie-ECO commended limits. By considering the carbon sequestration of the wood products during their lifespan, the unit is very close to carbon neutrality. Overall, this work has demonstrated how a mixed-use unit as part of a multi-story building in a central European climate can achieve nZEB standards as well as outstanding energy and exergy performance, using vertical outdoor exposed surface only, equipped with advanced solar passive and active technologies. This approach may answer many challenges, identified in the cited literature, to the large-scale adoption of nZEB standards.

**Supplementary Materials:** The following are available online at www.mdpi.com/article/10.3390/en15030793/s1, Table S1: List of sensors within the NEST SolAce unit.

**Author Contributions:** Conceptualization, P.F., J.F. and A.S.; methodology, P.F., J.F. and A.S.; software, P.F., J.F., C.C. and X.T.; validation, P.F., J.F. and A.S.; resources, J.-L.S.; data curation, P.F. and J.F.; writing—original draft preparation, P.F., J.F. and C.C.; writing—review and editing, P.F., A.S. and J.-L.S.; visualization, P.F. and J.F.; supervision, A.S. and J.-L.S.; project administration, J.-L.S.; funding acquisition, J.-L.S. All authors have read and agreed to the published version of the manuscript.

**Funding:** This research was funded by Innosuisse—Schweizerische Agentur für Innovationsfördeung, as part of the Swiss Competence Center for Energy Research 'Future Energy Efficient Buildings and Districts' (SCCER FEEB & D).

**Institutional Review Board Statement:** Not applicable.

**Informed Consent Statement:** Not applicable.

**Data Availability Statement:** Data about NEST SolAce are property of EPFL and EMPA. Data about the NEST building and the Energy-hub are property of EMPA. Please refer to the official NEST website for data enquiries: https://www.empa.ch/web/nest (accessed on 20 January 2022).

**Acknowledgments:** This research project was financially supported by the Swiss Innovation Agency lnnosuisse as part of the Swiss Competence Center for Energy Research 'Future Energy Efficient Buildings and Districts' (SCCER FEEB & D). Special thanks are due to the EMPA NEST team in Dübendorf for their supportive and professional assistance. The authors would also like to thank Andrea Gasparella for the fruitful suggestions to enhance the quality of this work.

**Conflicts of Interest:** The authors declare no conflict of interest. The funders had no role in the design of the study; in the collection, analyses, or interpretation of data; in the writing of the manuscript, or in the decision to publish the results.

## Nomenclature

*Symbols*

| | | |
|---|---|---|
| $\theta_o$ | °C | Outdoor facing ceiling surface temperture |
| $\theta_e$ | °C | Outdoor air temperature |

| $\theta_i$ | °C | Indoor air temperature |
|---|---|---|
| $N_4$ | - | Outdoor air temperature correction coefficient (see EnergyPlus documentation [26]) |
| $N_7$ | - | Indoor air temperature correction coefficient (see EnergyPlus documentation [26]) |
| $P_{conduction}$ | W | Heat flux exchanged through the thermal envelope by conduction |
| $U_{tr}$ | W/(m² K) | U-value of the glazing part of the thermal envelope, from manufacturer data |
| $A_{tr}$ | m² | Total surface of the glazing part of the envelope |
| $P_{op}$ | W | Heat flux exchanged through the opaque part of the thermal envelope |
| $\Delta T$ | °C | Difference between indoor and outdoor air temperature |
| $P_{ventilation}$ | W | Heat flux exchanged by convection through ventilation |
| $D_{air}$ | m³/h | Volumetric airflow rate |
| $c_p$ | J/(kg K) | Air specific heat capacity |
| ρ | kg/m³ | Air density |
| $\eta_r$ | - | Heat recovery effectiveness of the ventilation system |
| $P_{solar}$ | W | Passive solar gains |
| $g$ | - | Solar heat gain coefficient |
| $\tau$ | - | Luminous transmission factor |
| $I_{GH,m}$ | W/m² | Global horizontal solar radiation issued from the local meteorologic station |
| $I_{GSW,TMY}$ | W/m² | Southwest vertical solar radiation issued from the typical meteorological year |
| $I_{GH,TMY}$ | W/m² | global horizontal solar radiation issued from the typical meteorological year |
| $P_{internal\ gain.}$ | W | Internal heat gains |
| $P_m$ | W | Metabolic rate of a person for a common office activity |
| $N$ | - | Number of people |
| $P_{elec}$ | W | Electric power absorbed by the electric appliance |
| $F$ | - | Ratio of electricity converted into heat |
| $Q_{heating}$ | kWh/year | Imported/exported heat for space heating |
| $Q_{DHW}$ | kWh/year | Imported/exported heat for domestic hot water heating |
| $Q_{cooling}$ | kWh/year | Imported/exported heat for space cooling |
| $Q_{PV}$ | kWh/year | Exported electricity issued from photovoltaic modules |
| $Q_{STC}$ | kWh/year | Exported heat from the solar thermal collectors |
| $Q_{elec}$ | kWh/year | Imported electricity for lighting and appliance |
| $w_{i,j,k,l,m,n}$ | - | Exergy-based, time-dependent weighting factor for energy fluxes |
| $m_{CO_2}$ | kg | Mass of sequestered $CO_2$ during the product lifespan |
| $m_{dry\ wood}$ | kg | Mass of dry wood constituting the product |
| $c$ | % | Fraction of dry mass represented by carbon |
| $f_{CO_2}$ | kg/kg | Amount of carbon dioxide cumulated in a unit of carbon mass |
| $n_{years}$ | - | Number of years assumed for the unit life cycle |

*Acronyms*

| BiPV | Building integrated photovoltaic |
|---|---|

| DGI | Daylight glare index |
| DHW | Domestic hot water |
| HBI | Human–building interaction |
| HDR | High dynamic range |
| HFA | Heated floor area |
| HTE | High temperature water network |
| HVAC | Heating ventilation air components |
| KBOB | Swiss coordination panel on construction issuing the Swiss database for sustainable construction (under the same acronym) |
| LCA | Life cycle assessment |
| MKZ | Primary energy index |
| MTE | Medium temperature water network |
| NEPF | Net energy performance figure |
| NTE | Cold temperature water network |
| nZEB | nearly nero-energy building(s) |
| NZEB | Net zero-energy building(s) |
| PCM | Phase change material(s) |
| PPD | Predicted percentage of dissatisfied |
| PV | Photovoltaic |
| SIA | Société Suisse d'Ingénieurs et Architectes (National standards organisation) |
| STC | Solar thermal collector(s) |
| *TMY* | Typical meteorological year |
| WWR | Windows-to-wall ratio |

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
