# Peer review of "Performance Assessment of a nZEB Carbon Neutral Living/Office Space and Its Integration into a District Energy-Hub"

_energies, doi:10.3390/en15030793_

Round 1

Reviewer 1 Report

The abstract is quite complete corresponds to the content of the article, contains links to the results of modern research. Dynamic simulations and on- site data monitoring were used to obtain a reliable energy balance in a form of Sankey diagrams. The article is developed and executed qualitatively, has a high scientific potential.

Author Response

The authors kindly thank Reviewer 1 for the appreciation.

Reviewer 2 Report

The authors have carried out the performance assessment of an nZEB Carbon Neutral Living/office Space and Its Integration Into a District Energy-hub. Its an interesting work. The paper has been written very well. Addressing the following comments could improve the manuscript further. 

1) It would be more understandable if the authors first illustrate the complete architecture of the research work through pictorial representation after the introduction section or in the appropriate place. 

2) Introduction section need to be elaborated. The authors need to provide more background on NZEB and bioclimatic buildings. 

3) Provide general pictorial representation of the NEST infrastructure/energy-hub in the Section 3. Further list the important components in the description of the unit. 

4) List the major findings and contributions of the work with respect to the state of the art systems. 

Author Response

The authors kindly thank Reviewer 2 for the contribution to improving the quality of the paper.

Here below we address (in red) the points raised by the reviewer individually:

1) It would be more understandable if the authors first illustrate the complete architecture of the research work through pictorial representation after the introduction section or in the appropriate place. 

A) A flowchart illustrating the main steps carried out during the research work is shown in Figure 1 in the revised manuscript.

2) Introduction section need to be elaborated. The authors need to provide more background on NZEB and bioclimatic buildings. 

B) A paragraph with the definition of nZEB has been incorporated in the introduction (lines 45-49 of the revised manuscript), to better specify what it is meant.

3) Provide general pictorial representation of the NEST infrastructure/energy-hub in the Section 3. Further list the important components in the description of the unit. 

C) A new Figure 2 f) is incorporated in the revised manuscript to provide a pictorial representation of the Energy-hub within the NEST infrastructure.

D) Table 1 is incorporated in the manuscript, listing the most significant and innovative components featured in the unit.

4) List the major findings and contributions of the work with respect to the state of the art systems. 

E) A dedicated paragraph (lines 603-608 of the revised manuscript) has been added to the conclusion section to stress the achievements.

Reviewer 3 Report

The manuscript presents the energy performance assessment of a nearly zero energy building integrated into a district energy hub in Dubendorf Switzerland. In general, the article falls within the aim and scope of Energies Journal. In addition to that, the paper is well-written and structured, and the provided results are interesting to the energy consultants and the policy makers in the field. Thus, I recommend this work for publishing.

Author Response

Authors kindly thank Reviewer 3 for the appreciation. A thorough review of English expression has been carried out the the best of our capabilties.

Reviewer 4 Report

The paper is well-written with a good explanation of the data-monitoring and the hybrid model. I suggest publishing it after considering very minor points as follows: 

  • It is suggested to clarify how the uncertainty level is calculated for data-driven model in line 364. 
  • It is suggested to replace low-quality figures. 

Author Response

The authors thank Reviewer 4 for the valuable contribution to improving the quality of the work.

We address hereafter in red the comments made by the reviewer:

  • It is suggested to clarify how the uncertainty level is calculated for data-driven model in line 364. 
  • With uncertainty we mean here the discrepancy between input and output energy in the Sankey diagrams, which creates a gap at the centre of the diagrams. We rephrased the sentence to clarify (line 379 of the revised manuscript).
  • It is suggested to replace low-quality figures.
  • Best available resolution figures have been employed in the manuscript.